# Environmental Safety Assessments of Lipid Nanoparticles Loaded with Lambda-Cyhalothrin

**DOI:** 10.3390/nano12152576

**Published:** 2022-07-27

**Authors:** Catarina Ganilho, Márcia Bessa da Silva, Cristiana Paiva, Thacilla Ingrid de Menezes, Mayara Roncaglia dos Santos, Carlos M. Pereira, Ruth Pereira, Tatiana Andreani

**Affiliations:** 1GreenUPorto, Sustainable Agrifood Production Research Centre & INOV4AGRO, Department of Biology, Faculty of Sciences, University of Porto, Rua Campo Alegre s/n, 4169-007 Porto, Portugal; up201707255@edu.fc.up.pt (C.G.); bessamiss@gmail.com (M.B.d.S.); cristiana.pgpaiva@fc.up.pt (C.P.); 2Chemistry Research Centre (CIQUP) & Institute of Molecular Sciences (IMS), Department of Chemistry and Biochemistry, Faculty of Sciences, University of Porto, Rua do Campo Alegre s/n, 4169-007 Porto, Portugal; up201811986@edu.fc.up.pt (T.I.d.M.); mayara.santos@fc.up.pt (M.R.d.S.); cmpereir@fc.up.pt (C.M.P.); 3Centre for Research and Technology of Agro-Environmental and Biological Sciences (CTAB) & INOV4AGRO, University of Trás-os-Montes e Alto Douro, UTAD, 5000-801 Vila Real, Portugal

**Keywords:** nanopesticides, insecticides, ecotoxicology, non-target terrestrial organisms, soil microbial parameters

## Abstract

Lipid nanoparticles (LN) composed of biodegradable lipids and produced by green methods are candidates for the encapsulation of pesticides, potentially contributing to decreasing their release in the environment. From a safety-by-design concept, this work proposes LN for the encapsulation of insecticide active ingredients (AI). However, given the complexity of nanoparticles, ecotoxicological studies are often controversial, and a detailed investigation of their effects on the environment is required. Accordingly, this work aimed to produce and characterize LN containing the insecticide lambda-cyhalothrin (LC) and evaluate their safety to crops (*Solanum lycopersicum* and *Zea mays*), soil invertebrates (*Folsomia candida* and *Eisenia fetida*), and soil microbial parameters. The average particle size for LN-loaded with LC (LN–LC) was 165.4 ± 2.34 nm, with narrow size distribution and negative charge (−38.7 ± 0.954 mV). LN were able to encapsulate LC with an entrapment efficacy of 98.44 ± 0.04%, maintaining the stability for at least 4 months. The LN–LC showed no risk to the growth of crops and reproduction of the invertebrates. The effect on microbial parameters showed that the activity of certain soil microbial parameters can be inhibited or stimulated by the presence of LN at highest concentrations, probably by changing the pH of soil or by the intrinsic properties of LN.

## 1. Introduction

Modern agriculture practices have been suffering constant pressure to increase food production, due to the rapid expansion of global population density, as well as the need to respond to highly demanding markets and consumers. The intensification of the use of agrochemicals, especially insecticides for pest control, has been one of the major consequences [1] and is partly responsible for different environmental issues, such as soil, water, and food contamination; pest resistance; bioaccumulation of pesticide residues in food chains; and decrease of biodiversity [2]. During the field application, only a fraction of the active ingredient (AI) of pesticide formulations reaches the desired target. The largest amount is lost because of improper pulverization practices and precipitation washing, being decomposed and/or becoming mobile in the environment through volatilization, soil percolation, and runoffs. Variable environmental conditions also affect pesticides’ stability [3], thus increasing the amounts that have to be applied [4] and the costs to the farmers. In most cases, and depending on its chemical properties, the agricultural ecosystems are unable to completely neutralize the applied insecticides, resulting in the accumulation of its residues in the non-target compartments, such as biota [5].

In recent years, different strategies have been proposed for the sustainable control of agricultural pests and diseases. In this context, nanoscience and nanotechnology developments have been targeting the agri-food sector through the design of nanopesticides (NPest) formulations [6], in which, for example, pesticide’s active ingredients (AI) are encapsulated in nanomaterials (NM) [7], with a size between 1 and 1000 nm [8]. The use of NPest has been announced as a promising strategy for pest management, since nanoformulations decrease the amount of pesticide AI needed by improving its stability (protecting AI from external agents, such as temperature, light, and moisture) and by controlling its targeted release in the environment with minimal environmental and human impact. In this context, lipid nanoparticles (LN) appeared as promising nanocarriers for insecticide AI loading, as they are composed of biocompatible and biodegradable lipids [3], have good physicochemical stability, promote sustainable release of AI, and their synthesis offers competitive costs and ease of manufacturing [9]. In addition, LN can be produced by ecofriendly methods, without the use of organic solvents [10]. However, only one study described the application of LN for the encapsulation of insecticide AI [11]. Furthermore, these studies only evaluate the efficacy of insecticide-loaded LN on target species, and, thus, there is still limited information about the safety of these systems to non-target organisms, specially to terrestrial species.

In this context, the present work aimed at encapsulating the insecticide lambda-cyhalothrin (LC) in LN, using physiological lipids and ecofriendly methods, as well as at evaluating their safety on agriculture crops (*Solanum lycopersicum* and *Zea mays*), on soil invertebrates (*Folsomia candida* and *Eisenia fetida*), and on soil microbial parameters.

Lambda-cyhalothrin (LC) is a type II pyrethroid insecticide [12] that mimics natural molecules extracted from chrysanthemum flowers (*Chrysanthemum cinerariaefolium* and *C. cineum*) with biocidal effect and therefore has been widely used to control insect pests in agriculture, veterinary, and domestic applications. However, after spraying products containing pyrethroids, their residues can be spilled on soil, where they can accumulate, as pyrethroids are highly hydrophobic compounds, binding to organic matter and other soil components [13]. Nevertheless, due to it risk to aquatic organisms and non-target arthropods, LC has been banned from the European market since 2021, based on the risk assessment of LC presented by The European Food and Safety Authority [14]. Thus, the encapsulation of LC in physiological and biodegradable lipids may be an excellent strategy to reduce its harmful effects on the environment and to recycle the use of the banned pesticides with a more sustainable perspective. In addition, the gathered data in the present study can be used in a safety-by-design (SbD) approach to select the concentrations of the encapsulated AI that are less harmful to non-target species, and they can then be tested for their efficacy to target species. To the best of our knowledge, the environmental safety of LC encapsulated in LN has not been previously reported, and, therefore, these findings can facilitate the understanding of the impact of encapsulated LC on the environment.

## 2. Materials and Methods

### 2.1. Materials

The AI lambda-cyhalothrin (LC) was purchased from LGC Standards (Barcelona, Spain). Precirol^®^ ATO 5 (Prec) (Glyceryl palmitostearate) (Gattefosse S.A., Saint-Priest, France) was used as a solid lipid, and Capryol 90^®^ (Propylene glycol monocaprylate) was used as a liquid lipid (Gattefosse S.A., Saint-Priest, France). Soy lecithin, used as amphipathic surfactant, was obtained from VWR (Alfragide, Portugal). The surfactant TegoCare 450^®^ (TG450) was supplied by Evonik Industries (Essen, Germany). Dimethyl sulfoxide (DMSO) was used as a dispersant agent to suspend LC and was obtained from Sigma-Aldrich (Lisbon, Portugal).

#### Preparation and Characterization of Lipid Nanoparticles (LN)

LN were prepared by a modified oil-in-water (O/W) emulsion method without the addition of organic solvents, as described previously [15]. Briefly, 0.1% (*w/w*) of LC was added to the oil phase composed of 3% (*w/w*) of glyceryl palmitostearate, 0.5% (*w/w*) of soy lecithin, 2% (*w/w*) of Capryol 90, and 1.5 % (*w/w*) of TG450 and heated at 5–10 °C above the melting point of the solid lipid (~65 °C). The aqueous phase was heated to the same temperature and then added to the oil phase, followed by high-speed homogenization at 15,000 rpm (Ultra-Turrax T25, IKA Labortechnik, Deutschland, Germany) for 10 min, with a probe (18 G). After homogenization, the dispersion was placed in an ice bath for 10 min for recrystallization of the lipid matrix and formation of LN.

LN were characterized for the average particle size, particle distribution expressed as polydispersity index (PI), and zeta potential (ZP). The average size and PI were determined by dynamic light scattering (DLS) with a Zeta Sizer NanoZS equipment (Malvern Panalytical, Malvern, UK), using disposable polystyrene cells, 173° scattering angle, and refractive index of 1.6. The ZP analysis of LN was performed in the same equipment by electrophoretic light scattering, using disposable plain folded capillary zeta cells. All measurements were performed at 25 °C, and the LN dispersions were diluted in deionized water in a 1:100 (*v/v*) ratio and analyzed in triplicate. The values were expressed as mean values ± standard deviation (SD). After production, unloaded and loaded LN were stored in the dark at 25 °C, and the variation of average particle size, PI, and ZP were recorded at predetermined time intervals (day 0 and 30, 60, and 120 days after LN synthesis).

Glyceryl palmitostearate in LN was indirectly measured by the filtration/centrifugation method, followed by the determination of the free amount of LC (non-encapsulated) by high-performance liquid chromatography (HPLC). A volume of 250 µL of LC-loaded LN dissolved in ethanol (1:1) was centrifuged for 10 min at 13,500 rpm, at 25 °C, in a centrifugal filter device (Amicon Ultra 0.5, NMWL 30 K, Merck Millipore Ltd., Dublin, Ireland). Then 100 µL of filtrate was diluted with ethanol at 1:4, and the concentration of LC was analyzed by HPLC. The chromatographic analysis was performed by Shimadzu Nexera-I-LC2040C-3D (Shimadzu, Kioto, Japan) equipped with Inertsil ODS 3V (150 mm × 4.6 mm, 3 µm) column (GL Science Inc., Tokio, Japan) at 30 °C and a UV–Vis detector set at 240 nm. The mobile phase consisted of acetonitrile:water (85:15, *v/v*). The flowrate was set to 0.1 mL/min, and the injection volume was 10 μL. The area under the curve was measured for the calculation of the LC concentration based on the calibration curve. The EE% was calculated by using the following equation:$$\%\;\mathrm{EE}=\frac{\mathrm{Total}\;\mathrm{amount}\;\mathrm{of}\;\mathrm{LC}\;\mathrm{AI}-\mathrm{Free}\;\mathrm{AI}\;}{\mathrm{Total}\;\mathrm{amount}\;\mathrm{of}\;\mathrm{AI}}\times 100$$

### 2.2. Environmental Safety Assessment of LN–LC on Terrestrial Organisms

#### 2.2.1. Test Soil

Natural soil used in the present study was collected from the 0–20 cm topsoil layer in the region of Vairão (Vila do Conde, Portugal), in the open fields of the GreenUPorto Research Centre, with no historical application of pesticides at least in the last three decades, sieved through a 4 mm mesh size and air dried. The soil physicochemical characterization presented in Table 1 was performed in the laboratory with soil sieved at 2 mm. Five replicates were used for all the following parameters: pH (H_2_O and KCl (1 M)), conductivity, organic matter content (%OM), and water-holding capacity (% WHCmax). The pH and conductivity were determined in a 1:5 (*v/v*) soil suspension. After 60 ± 10 min of agitation, the suspension was allowed to stand for 60 min before measuring pH with a pre-calibrated multiparameter (Edge, HANNA Instruments, Povoa de Varzim, Portugal) [16]. Conductivity was recorded with the pre-calibrated meter on the H_2_O suspension [17]. For OM content, soil samples were dried at 105 °C until constant weight and then obtained by loss on ignition of dried soil samples at 450 °C, for 8 h, and expressed in percentage [18]. For WHC_max_ (%), soil samples were introduced in polypropylene containers whose bottom was replaced with filter paper and immersed in water for 3 h. Subsequently, soil samples were drained with successive exchanges of absorbent paper for 2 h and dried at 105 °C [19].

#### 2.2.2. Assessment LN–LC Safety to Terrestrial Plants Growth

For the plant growth assay, two species were used: one dicotyledonous plant, *Solanum lycopersicum* (tomato); and one monocotyledonous plant, *Zea mays* (corn). Seeds were purchased from a local supplier in the city of Porto and used for the assay, following the standard protocol OECD 227 [20]. For this purpose, 200 g of natural soil was added to plastic pots moistened to the WHC_max_ of the natural soil (~43%). For *S. lycopersicum* and *Z. mays*, we added 20 and 10 seeds to each pot, respectively. The test started after the germination of 70% of the seeds. Then 10 seedlings for *S. lycopersicum* and 7 seedlings for *Z. mays* were kept in each pot, thus avoiding intraspecific competition. The plants germinated and grew in a growth chamber with a controlled temperature (20 ± 2 °C), a photoperiod (16 h^L^:8 h^D^), and photosynthetically active radiation (25,000 lux). After 15 days of growth and formation of the 2nd and 3rd true leaf, the plants were sprayed on the shoots with a volume of 5 mL of the nanoformulation containing LC at concentrations of 0, 7, 10, 14, 20, and 28 g LC ha^−1^ (based on the dose of LC recommended in the commercial formulation Karate+ (Syngenta)). The compounds tested were LC AI dispersed in DMSO at 5% (*v/v*) and LC-loaded in lipid nanoparticles (LN) in an aqueous dispersion (LN–LC). The effect of DMSO at 5% (*v/v*) was also tested in a DMSO control. The unloaded LN were not tested in this case because the LC-loaded LN did not cause significant effects. For the plants’ growth assays, 5 replicates were tested for the control group and for each compound tested concentration. In the control, plants were only sprayed with deionized water. After 10 days of exposure, plants from each replicate were harvested, and their fresh and dry biomass (roots and shoots) and the length of the roots were evaluated.

#### 2.2.3. Assessments of LN–LC Safety to Soil Invertebrates

Since the interaction between the AI encapsulated into LN and the biological membranes can be mediated by the release profile of AI from nanoparticles, the short- and long-term effects of nanoformulations were evaluated on the avoidance, survival, and reproduction of the earthworm *Eisenia fetida* (Oligochaeta: Lumbricidae) and the collembola *Folsomia candida* (Collembola: Isotomidae). All organisms were obtained from laboratorial cultures kept under controlled conditions (temperature, 20 ± 2 °C; photoperiod, 16 h^L^:8 h^D^). The organisms were fed weekly. Test organisms are collected from synchronized cultures with homogeneous age.

##### Reproduction Test with *F. candida*

The reproduction test with *F. candida* species was performed according to the OECD standard protocol 232 [21]. The assay was performed in small plastic containers containing 30 g of natural soil thoroughly mixed with the nanoformulations at the concentrations 0, 7, 10, 14, 20, and 28 g LC ha^−1^ (based on the doses of LC recommended in the commercial formulation). The formulations tested were LC AI solution (LC dispersed in DMSO at 5% (*v/v*)), unloaded LN, and LC-loaded LN (LN–LC). The concentrations of LN without LC were based on the amount of solid lipid (SL) used in the synthesis ranging from 0, 150, 210, 300, 480, and 660 g SL ha^−1^. The concentrations of all LN constituents (lipids and surfactants) were the same for unloaded and loaded LN with LC. The effect of DMSO at 5% (*v/v*) was also tested in a DMSO control. The soil WHC_max_ was adjusted to 50%. The water used to adjust soil moisture was used to add the formulations at the concentrations mentioned. Thereafter, ten individuals (9–12 days old) were placed in the plastic containers, and dry yeast was added as food. Each concentration and the control groups were tested in five replicates. After 28 days of exposure, under the same conditions as described for the cultures, the plastic containers were filled with water, gently mixed with the soil, and then transferred to larger plastic pots. Afterward, few drops of China ink were added and carefully homogenized. The collembola floated on the water surface, which was photographed to allow us to count the number of the adults and juveniles in each pot, using a public domain software ImageJ version 1.53k (Wayne Rasband and contributors, National Institutes of Health, Maryland, USA – Java 1.8.0_172 (64-bit) – http://imagej.nih.gov/ij).

##### Avoidance and Reproduction Tests with *E. fetida*

Avoidance tests with *E. fetida* were carried out in plastic boxes and followed the standard protocol ISO 17512-1 [22]. The boxes were divided into two equal compartments with a paperboard divider. To each compartment, 200 g of natural soil were added with the soil WHC_max_ adjusted to 50%. The water used to adjust soil moisture was used to add the formulations at the concentrations mentioned. On the left side, we added non-contaminated soil (control), while on the right side, we placed soil spiked with different concentrations of unloaded LN and loaded LN with LC (LN–LC) dispersions, as well as the AI LC. The concentrations tested range included 0, 7, 10, 14, 20, and 28 g LC ha^−1^ (based on the dose of LC recommended in the commercial formulation Karate+, Syngenta), and the concentrations of unloaded LN were based on the amount of solid lipid (SL) used in the synthesis range that included 0, 150, 210, 300, 480, and 660 g SL ha^−1^. The concentrations of all LN constituents (lipids and surfactants) were the same for unloaded and loaded LN with LC. The control group was moistened only with deionized water and tested in five replicates, while four replicates were tested per concentration for the contaminated soils. After preparing all replicates, 10 adult earthworms (300–600 mg) were placed between both compartments. Boxes were covered with a perforated cap and maintained for 48 h in the same conditions (Section 2.2.3), without adding food. After 48 h of exposure, the number of earthworms on each side (control and treatments) of the boxes was counted, and the average avoidance percentage was calculated.

The reproduction test with *E. fetida* followed the standard OECD 222 protocol [23]. Before the assay, clitellate adult oligochaetes (300–600 mg) were acclimatized in the natural soil used in this study, for 48 h, under the same culture maintenance conditions (Section 2.2.3). The test was conducted in plastic containers with 500 g of natural soil homogenized, with deionized water (control group) and with the nanoformulation at the concentrations 0, 7, 10, 14, 20, and 28 g LC ha^−1^, based on the dose of LC recommended in the commercial formulation. The formulations tested were LC AI dispersed in DMSO at 5% (*v/v*), unloaded LN, and LC-loaded LN (LN–LC) dispersions. The concentrations of unloaded LN were based on the amount of solid lipid (SL) used in the synthesis range that included 0, 150, 210, 300, 480, and 660 g SL ha^−1^. The concentrations of all LN constituents (lipids and surfactants) were the same for unloaded and loaded LN. The effect of DMSO at 5% (*v/v*) was also tested. The soil WHC_max_ was adjusted to 50%. The water used to adjust soil moisture was used to add the formulations at the concentrations mentioned. Ten oligochaetes with clitellum were added to each container and kept for 56 days. Each concentration was tested in quadruplicate, while the control group was tested in quintuplicate. After 28 days of exposure, the adults were removed and counted, and the test was extended for a further 28 days to allow the juveniles to grow. At the end of the test (56 days), the juveniles were extracted from the soil with heat, in a water bath (60 °C for 15 min), and the number of juveniles per replicate was counted.

#### 2.2.4. Assessments of LN–LC Safety to Soil Microbial Parameters

The collected natural soil was sieved at 2 mm. The stock solution of the formulations was diluted at different concentrations in the amount of water required to adjust the WHCmax to 80% and maintained for 15 days, at 20 ± 2 °C, and for a photoperiod of 16h^L^:8h^D^. The moisture of the control group was adjusted only with deionized water. During this period, the soil was weighted every 3 days, and the moisture was adjusted with deionized water when necessary. After 15 days of exposure, for each replicate (both control and contaminated soils), 3 sub-replicates were prepared by transferring 1 g of soil to individual Falcon tubes, which were stored at −20 °C for a maximum of 1 month.

The pH variation in soil due to the presence of contaminants can affect the behavior of the soil microbial community and, consequently, its biological activity. Thus, after incubation, the pH of the control soil and of the soils treated with the highest concentrations tested was again checked in a suspension in KCl (1 M) [16].

For the assessment of soil microbial parameters, changes in the activity of the enzymes dehydrogenase, acid phosphatase, arylsulfatase, carboximetil cellulase (CM-cellulase), and urease were measured. In addition, the potential nitrification and nitrogen mineralization were also evaluated.

For the analysis of dehydrogenase activity [24], soil samples were incubated at 40 °C, for 24 h, with a solution of triphenyltetrazolium chloride (TTC) (10g L^−1^) in Tris buffer 0.1 M at a pH of 7.6, while the blank tubes were incubated only with Tris buffer 0.1 M at a pH of 7.6. The triphenylformazan (TPF), which is produced by the reduction of TTC, was extracted with acetone, forming a pink-colored complex. The absorbance was then measured spectrophotometrically at 546 nm. The dehydrogenase enzymes’ activity was determined from a standard curve obtained for TPF in acetone and expressed in µg TPF g^−1^dm (% dry matter) h^−1^.

For the acid phosphatase enzyme [25], soil samples were incubated at 37 °C for 1 h with p-nitrophenylphosphate solution (115 mM) in standard buffer at pH 6.5. The blank tubes were incubated only with standard buffer at a pH of 6.5. The p-nitrophenol (pNP) released by the phosphomonoesterase activity was extracted with sodium hydroxide (0.5 M), forming a yellow complex, which was measured at 405 nm. The acid phosphatase enzyme activity was determined from a standard curve and expressed in µg pNP g^−1^dm h^−1^.

For arylsulfatase enzyme analysis [26], soil samples were incubated at 37 °C for 1 h with a p-nitrophenylsulfate solution (0.02 M) in acetate buffer 0.5 M at a pH of 5.8. The blank tubes were incubated only with sodium acetate trihydrate buffer (0.5 M) at a pH of 5.8. The nitrophenol (pNP) released by the arylsulfatase activity was extracted by NaOH 0.5 M resulting in a yellow-colored complex, which was measured at 420 nm. The activity of the arylsulfatase enzyme was determined from a standard curve and expressed in µg pNP g^−1^dm h^−1^.

For the analysis of the carboximetil cellulase (CM-cellulase) [27], using carboxymethyl cellulose as the substrate (0.7% *w/v*), the soil samples were incubated for 24 h at 50 °C and a pH of 5.5. The blank tubes were incubated only with acetate buffer (2M) at a pH of 5.5. The reducing sugars produced during the incubation period caused the reduction of potassium hexacyanoferrate (III) to potassium hexacyanoferrate (II) in an alkaline solution. The latter compound reacted with ferric ammonium sulfate in an acid solution to form a blue-colored ferric hexacyanoferrate (II) complex, which was measured at 690 nm. The activity of the CM-cellulase enzyme was calculated from a standard curve obtained for defined concentrations of glucose, in aqueous solution, and expressed in µg glucose (GLU) g^−1^dm 24 h^−1^.

For the analysis of the urease enzyme activity [28], the soil samples were incubated at 37 °C, for 2 h, with a borate buffer 0.1 M at pH 10 and a solution of urea (720 mM) in the same buffer. The blank tubes were incubated only with borate buffer solution (0.1 M; pH 10). The ammonia released by the enzyme activity was extracted with KCl 2 M, forming a green-colored complex, which was measured at 690 nm. The activity of the urease enzyme was calculated from a standard curve, using defined concentrations of NH_4_^+^, in aqueous solution, and expressed in µg NH_4_^+^ g^−1^dm 2h^−1^.

For the evaluation of the nitrification potential [29], soil samples were incubated at 25 °C, under in an orbital shaker, and the blanks were incubated at −20 °C for 5 h, with ammonium sulfate (NH_4_)_2_SO_4_ 1 mM as the substrate and NaCl 1.5 M. The nitrite released by the enzyme activity was extracted with KCl 2 M, forming a pink-colored complex, which was measured at 502 nm. The nitrification was determined from a standard curve, using defined concentrations of NO_2_ in aqueous solution, and expressed in ng of nitrite NO_2_^−^ g^−1^dm 5h^−1^.

For the evaluation of nitrogen mineralization [30], soil samples and the blanks were incubated at 40 °C for 7 days, with deionized water. In this process, the organic forms of N were metabolized into inorganic N forms, such as NH_4_^+^, which were determined after the extraction with KCl 2 M. After the reaction of ammonia with sodium salicylate and sodium nitroprusside in the presence of sodium dichloroisocyanurate, a green-colored complex was formed and measured spectrophotometrically at 690 nm. The mineralization of nitrogen was determined from a standard curve, using defined concentrations of NH_4_^+^, in aqueous solution, and expressed as µg NH_4_^+^ g^−1^dm d^−1^.

### 2.3. Statistical Analysis

All statistical procedures were performed by using Prism version 8.0.2 (263), created by GraphPad Software (San Diego, California, USA). For all the endpoints, data were expressed as mean values ± standard deviation (SD). Regarding hypothesis testing, the data obtained for the average particle size, PI, and ZP of nanoformulations were tested for significant differences between storage periods (30, 90, and 120 days) and time 0 (day of the production) for LN or LN–LC and between unloaded LN and loaded LN–LC, within the same storage period by analysis of variance (two-way ANOVA), with subsequent post hoc comparison, using Tukey’s test. The analysis of the effects on plants growth endpoints (*S. lycopersicum* and *Z. mays*), on reproduction of the soil invertebrates (*E. fetida* and *F. candida*), and on soil microbial parameters was performed by analysis of variance (ANOVA), followed by a Dunnett multi-comparison test to check for differences from the control group (CTL). Whenever ANOVA assumptions were not met, a non-parametric Kruskal–Wallis test, followed by a Dunn’s multi-comparison test, was performed. For the avoidance test with *E. fetida*, Fisher’s exact test was performed by using GraphPad online software (https://www.graphpad.com/quickcalcs/contingency1.cfm, accessed on: 20 June 2022) to test for no avoidance between soils. The variations in the pH value were calculated by analysis of variance (ANOVA), followed by a Dunnett multi-comparison test to check for differences with the control group (CTL). Statistically significant differences were considered for a significance level α = 0.05.

## 3. Results and Discussions

### 3.1. Physicochemical Characterization of LN

Synthesized LN were evaluated with respect to their average particle size (nm), PI, and ZP (mV), which are essential physicochemical parameters that are necessary in order to obtain information concerning the appearance, consistency, and stability of colloidal systems [31]. The measurements of unloaded and loaded LN (LN–LC) were performed immediately after production and 30, 60, and 120 days of storage in the dark, at 25 °C (Figure 1).

Initially, the average particle size values of unloaded LN were 175.4 ± 2.6 nm and showed no significant variation over the period (120 days) (*p* > 0.05) (Figure 1A). The encapsulation of LC in LN resulted in a significant decrease (*p* < 0.05; F (9, 18) = 91.50) in the average particle size to165.4 ± 2.343 nm; this may indicate that LC, by having a high hydrophobic character, has great affinity with the lipid matrix, thus providing a greater homogenization of all constituents that compose the LN structure (Figure 1A). Similarly, to unloaded LN, the average particle size of LN–LC remains in the same range for 120 days, thus indicating no tendency to form aggregation and the high stability of LN after LC encapsulation when stored at 25 °C.

Regarding particle size distribution, immediately after production (day 0), unloaded LN formulations showed low PI values (0.293 ± 0.006) and were not significantly affected after LC encapsulation (PI = 0.264 ± 0.012) (*p* > 0.05; F (9, 18) = 10.19) (Figure 1B). Over the 120 days, the PI values decreased significantly for both formulations LN and LN–LC, when compared with day 0 (*p* < 0.05; F (9, 20) = 10.12) (Figure 1B)). The PI values of formulations were all lower than 0.3, which indicates that LN formed homogeneous suspensions [32]. The ZP values represent the electrical charge at the nanoparticle surface, as well as the degree of repulsion between similarly charged particles which avoids the occurrence of particle aggregation [33]. After synthesis, the ZP values of unloaded LN were −50 ± 1.52 mV (Figure 1C). However, after LC encapsulation, the ZP values decreased to −38.7 ± 0.954 mV (Figure 1C), which indicates that an amount of the insecticide could be located on the LN surface, and thus influencing the surface charge of nanoparticles (*p* < 0.05; F (9, 20) = 70.99). Nevertheless, In the present study, all ZP values recorded were above |30 mv|, which is an indicator of higher electrostatic repulsion between particles, providing a greater colloidal stability over time [34].

The interaction of LC and LN was investigated by using the evaluation of entrapment efficiency (EE). Since LC is considered a poorly aqueous soluble compound (solubility in water = 0.005 mg L^−1^ at 20 °C), LN are suitable nanocarriers for LC encapsulation, showing a %EE of 98.44 ± 0.04. The high %EE can be attributed to the reduced particle size of synthesized LN and to the presence of liquid lipid (Capryol 90) in the LN structure, as this can favor the solubility of LC in the melted lipid matrix, as well as the formation of LN with a more amorphous structure to improve the LC entrapment. Thus, the synthesized LN showed good properties regarding average particle size, homogeneity, and surface charge, demonstrating that LN are excellent systems for LC encapsulation. 

### 3.2. Assessment of LN Environmental Safety

NPest have been widely developed in recent years. However, knowledge about their fate and environmental effects is very limited, and it is still unclear whether NPest will result in significant benefits over conventional products. From the point of view of study design and correct interpretation of results, the data relating to the safety of developed NPest is still unclear, and, thus, this present study tries to fill this gap, at least for LC-loaded LN.

The effect of nanoformulations loaded with LC was evaluated by using a monocotyledonous plant, *Z. mays*, and a dicotyledonous plant, *S. lycopersicum*. According to the results, no statistically significant differences were recorded on the fresh biomass of the shoots (*p* > 0.05; F (12, 50) = 1.210) and roots (*p* > 0.05); F (12, 50) = 0.9060) of *S. lycopersicum* (Figure 2a,b); moreover, no significant differences were recorded on the fresh biomass of the shoots (*p* > 0.05) and roots (*p* > 0.05; F (12, 50) = 1.230) of *Z. mays* (Figure 3a,b) after exposure to increasing concentrations of LC and LN–LC by foliar application when compared to the control group. Thus, LN–LC demonstrated itself to be safe for the plants selected in this study. However, LC negatively affected the dry biomass of the roots of *S. lycopersicum* (LOEC = 28 g LC ha^−1^, NOEC = 20 g LC ha^−1^; *p* < 0.05; Kruskal–Wallis statistics value = 21.44), as indicated in Figure 2d, while LC stimulated *Z. mays* root length at 20 g LC ha^−1^ (*p* < 0.05; F (12, 302) = 5.01) (Figure 3e). No statistically differences were recorded for both species after exposure to DMSO at 5% (*v/v*) in comparison with the control group (data not shown).

Plants’ growth and development depend on their adaptation to a constantly changing abiotic environment, as well as their contact with pesticide residues present in soil [35]. Although there are several works about the effect of nanomaterials on plants, the most are related to metallic/inorganic nanomaterials [36,37], and, to the best of our knowledge, no data on LN phytotoxicity exist.

Concerning the LC effects on terrestrial plants, the data are also scarce and depend on the plant species. In our study, dicotyledonous species appeared to be more sensitive than monocotyledonous species were to LC exposure, and, in the same way, the development of roots was more affected than the that of the shoots. However, a comparative study conducted by Bragança et al. (2018) to evaluate the effects of pyrethroids (0 to 500 µg kg^−1^_soil_) on *Cucumis sativa* showed that LC was not toxic to plants, while cypermethrin decreased the shoot length (50 to 500 µg kg^−1^_soil_) compared to the control group [38]. This is in agreement with our results, since the maximum concentration tested for LC was 28 g LC ha^−1^ (which corresponded to 133 µg kg^−1^_soil_). On the contrary, Liu et al. (2009) showed that the application of the insecticide cypermethrin tested at concentrations ranging from 0 to 64 mg kg^−1^_soil_ (0–13 kg LC ha^−1^) reduced the root elongation (mm) of Pakchoi (Chinese cabbage) [39] [However, the maximum concentration of LC tested by these authors (13 kg LC ha^−1^) is much higher than those we tested and seemed to be well above recommended application doses.

Regarding the avoidance test with *E. fetida***,** the results revealed that oligochaetes did not avoid LC contaminated soil (Figure 4a). Contrarily, the earthworms significantly avoided LN–LC contaminated soil at the highest doses (20 and 28 g LC ha^−1^) (*p* < 0.05) (Figure 4c): LOEC 20 g LC ha^−1^ and NOEC 14 g LC ha^−1^) and significantly avoided LN contaminated soil at 150 and 480 g SL ha^−1^ (*p* < 0.05) (Figure 4b) compared to control soil. However, a maximum avoidance percentage of 75% was only recorded for the dose 20 g LC ha^−1^ when the active ingredient was encapsulated (LN–LC), which means that the habitat function of these soils was not compromised (<80% avoidance) even for the highest concentration tested (Figure 4c). Although the avoidance behavior is usually interpreted as a negative response, the results obtained in our study allow to conclude that these invertebrates may be less exposed to LC when it is encapsulated in LN, as somehow LN may be promoting an avoidance behavior, letting them to escape from a higher exposure level.

In addition, these findings may indicate that, despite detecting the presence of LC, the earthworms behavior did not appear to be affected by the concentrations tested or these concentrations may have inhibited the response ability of the organisms to the insecticide, leading to muscle paralysis, inconsistent movements, or the absence of sensorial detection [40].

Despite the avoidance results, the reproduction tests with *E. fetida* were not affected by the presence of LN (*p* > 0.05; F (5, 19) = 1.054) and LN–LC (*p* > 0.05; F (5, 19) = 1.643) after 56 days of exposure to increasing concentrations when compared to the control group (Figure 5b,c), and no statistical differences were recorded after exposure to DMSO at 5% (*v/v*) (data not shown). Although there are no data regarding the effect of LN on oligochaetes, for comparison, and despite being a work where a surfactant was tested, a study conducted by Gavina et al. (2016) showed that the earthworms *Eisenia andrei* significantly avoided the contaminated soil with nanovesicles made from the anionic surfactant sodium dodecyl sulphate and the cationic lipid didodecyl dimethylammonium bromide (SDS/DDAB), while the reproduction of the same invertebrates was not affected, probably by the rapid degradation of the vesicles in the soil [41]. Since the nanoparticles in the present study were produced by using lipids, it is possible that, initially, the earthworms were affected by their presence, but along the time, these LN can be (bio)degraded, and the soil habitat function can be restored. The reproduction of *E. fetida* was not also significantly affected by LC as shown in Figure 5a and the survival of adult earthworms was not affected when exposed to LN and LC (data not shown) at the concentrations tested. However, there are studies that state that LC is very toxic to *E. fetida* due to the high cutaneous absorption of these compounds by earthworms (LC_50_ from 1055 to 2570 g ha^−1^) [42], but these concentrations are higher than those tested in our study, which are likely more ecologically relevant, as they are close to the application dose.

According to the results, the reproduction of *F. candida* was significantly affected only by the presence of LC (*p* < 0.05; F (5, 24) = 5.903) at all the concentrations tested, as illustrated in Figure 6a (NOEC < 7g LC ha^−1^, LOEC ≤ 7 g LC ha^−1^), while LN (*p* > 0.05; F (5, 24) = 0.7850) and LN–LC (*p* > 0.05; F (5, 24) = 0.8134) did not affect the reproduction of collembola (Figure 6b,c). In fact, the encapsulation of LC in LN decreased the toxicity of LC after 56 days of exposure to increasing concentrations of the insecticide (Figure 6c). The survival of adult springtails was not affected when exposed to LN and LC (data not shown), and as expected, the reproduction of *F. candida* was more sensitive than mortality to insecticide application. No statistical differences were recorded after the exposure of invertebrates to DMSO at 5% (*v/v*) (data not shown).

Soil microbiological and biochemical parameters are essential indicators of microorganism metabolism and function, nutrient cycling, and soil contamination [43]. Some studies have reported that the soil microbial community can be significantly compromised upon exposure to nanomaterials [44]. However, these studies were conducted to understand the impact on soil microbial community of metal/metal oxide nanomaterials [45] and polymeric nanoparticles [46].

In the present study, the effect of formulations was also evaluated in soil microbial activity through the analysis of enzymatic activities, N mineralization, and potential nitrification. As far as authors are aware, the present work is the first study gathering data concerning the effect of LC and LN on the soil microbial community. Since the soil physicochemical properties can influence the activity of soil enzymes, the pH was evaluated after the incubation of soil with different formulations, namely LC, LN, and LN–LC, and the results are presented in Table 2. The pH results demonstrated that the unloaded LN significantly increased the pH value of the soil (*p* < 0.05; F (2, 6) = 10.50), and the LC-loaded LN significantly decreased the pH of the soil (*p* < 0.05; F (2, 6) = 35.00).

A study has shown that pesticides exhibit inhibitory effects on dehydrogenase enzymes’ activity [46]. The dehydrogenases are involved in cellular respiration and are considered an indicator of overall microbial metabolic activity. Furthermore, the dehydrogenases have a relevant role in the oxidation of organic matter, with their activity being an excellent parameter for assessing the effect of NPest on the soil microbial community [47]. According to the obtained results, LC did not affect dehydrogenase activity for all concentrations tested (*p* > 0.05; F (10, 108) = 19.01) (Table 3 and Appendix A), while loaded LN with LC (*p* < 0.05; F (10, 24) = 16.34) significantly reduced the activity of dehydrogenase at highest concentrations compared to the control group (Table 3 and Appendix A). Unloaded LN significantly stimulated the activity at 210 and 300 g SL ha^−1^ and significantly inhibited the activity at the highest concentration tested (*p* < 0.05; F (10, 108) = 19.01) (Table 3 and Appendix A). Thus, our study suggests that, as nanoparticles are composed of lipids, at high concentrations, these lipids may promote the occlusion of soil pores [48]. The occlusion of soil pores can reduce soil oxygenation, decreasing the microbial population and, therefore, decreasing the activity of dehydrogenases at the highest concentrations tested (28 g LC ha^−1^/660 g SL ha^−1^) (Table 3 and Appendix A, respectively).

Regarding the enzymatic activity of CM-cellulase, it was found that the LC and unloaded LN did not significantly affect the enzyme activity, as shown in Table 3 and Appendix A, respectively (*p* > 0.05). However, after LC encapsulation in LN, an increase in this enzyme activity was observed being statistically significant at concentrations of 14 and 28 g LC ha^−1^ was observed, as indicated in Table 3 and Appendix A (*p* < 0.05; F (10, 24) = 5.925). The reason for an increase in the CM-cellulase activity after LN–LC could probably be due to the presence of the organic compound TegoCare 450, which can act as a glucose source since glucose is part of TegoCare constitution. Although this phenomenon was not observed for single LN, in some way, LC could have modified the LN structure, increasing the bioavailability of glucose from TegoCare. In other words, this would not be an increase in activity, but rather a quantification of glucose that did not result from the enzyme activity per se but rather from the LN themselves.

According to the enzymatic activity of urease, LN–LC significantly increased the enzyme activity compared to the control at 10, 14, and 20 g LC ha^−1^ (*p* < 0.05; Kruskal–Wallis statistics value = 68.29) (Table 3 and Appendix A). LN significantly increased the enzyme activity compared to the control at 480 and 660 g SL ha^−1^. However, the inhibition of urease enzyme in relation to LC (10 g LC ha^−1^) may have occurred by chance (*p* < 0.05; Kruskal–Wallis statistics value = 46.26) (Table 3 and Appendix A).

For arylsulfatase enzyme activity, it was observed that LN–LC caused a significant decrease in the activity compared to the control at the highest doses (20 and 28 g LC ha^−1^) (*p* < 0.05; F (10.124) = 4.161) (Table 3 and Appendix A). The reduction in activity caused by the dose of 20 g LC ha^−1^ is in line with the significant reduction caused by unloaded LN compared with the control (*p* < 0.05; F (10, 124) = 4.904) (Table 3 and Appendix A). The activity of arylsulfatases increases with the decreasing of S in soil [49], and their activity can be affected by the presence of contaminants, pH changes, and organic matter content [50]. As shown in Table 2, after the addition of LN–LC, there was a significant increase of pH when compared with the control soil, and, thus, the pH change—although small—favored by the presence of LN–LC could have affected the arylsulfatase activity.

However, so far, there are no scientific reports evidencing the effect of LC and LN on the enzymatic activity of arylsulfatase.

Regarding acid phosphatase enzyme activity, LC at doses of 14 to 28 g LC ha^−1^ (Table 3 and Appendix A) induced a significant reduction in enzyme activity compared to the control group. LN significantly decreased the enzyme activity compared to the control at all concentrations tested (Table 3 and Appendix A) (*p* < 0.05; Kruskal–Wallis statistics value = 71.23), while LN–LC caused a significant decrease only at the highest doses, as illustrated by Appendix A) (*p* < 0.05; Kruskal–Wallis statistics value = 39.41). Phosphatases are produced in soil when the concentration of phosphorus decreases and alkaline soil can favor the availability of this element [51]. According to Nannipieri et al. (1982), acid phosphatases can be inhibited by high concentrations of protons due to the ionization of specific groups of enzymes [52]. Therefore, since the soil pH is low (~4.61 ± 0.010 for LC), in the acidic medium, the amino groups of LC may be protonated (pka > 9), and the insecticide becomes positively charged [53]. For this reason, the inhibition of enzyme activity by LC at the highest concentrations can be related to the presence of protonated LC. In addition, and likely more relevant for explaining the inhibition of the acid phosphatase activity by LN and LN–LC, another phenomenon can be involved. LN are produced also by using soy lecithin that is a phospholipid, which could have been metabolized as a phosphorus source, thus increasing P element in soil and leading to an inhibition of acid phosphatase activity after the addition of LN.

The data obtained for nitrogen mineralization indicated that there were no inhibitory effects for all formulations tested, as shown in Figure 7. In opposition, LN at the highest concentration (*p* < 0.05; Kruskal–Wallis statistics value = 56.00) and LN–LC at intermediate concentrations significantly increased (*p* < 0.05; F (10, 124) = 12.54) the mineralization of nitrogen (Figure 7b,c). To date, there are no scientific reports that show the effect of LC and LN on nitrogen mineralization. However, these findings may indicate that this significant stimulation may have resulted from the composition of synthesized LN, since it has soy lecithin (rich in nitrogen) in its composition. When mineralization occurs, there is an increase in ammonia that will be converted to nitrites and nitrate through the nitrification process.

Although an increase of nitrogen mineralization was observed, LN–LC caused an inhibition of the potential nitrification, especially at the highest concentrations, as shown in Figure 8c (*p* < 0.05; F (10, 116) = 16.11). It is well-known that ammonia oxidizing bacteria are highly sensitive to changing environmental conditions, and, thus, it is possible that the changes in soil pH favored the inhibition of the enzymes responsible for the nitrification process (Table 2) [54]. Therefore, a decrease in pH values, although smaller, may have resulted in the inhibition of the potential nitrification.

The LN–LC affected the activity of the microbial community, especially the metabolism of nitrogen, sulfur, and phosphorus; and apart from the activity of phosphatases, the effects seem to result mainly from the LN and not from the insecticide LC. The effects observed do not indicate a negative impact on the microbial community, except for that suggested by the inhibition of dehydrogenase at the highest concentration; rather, they are likely indicators of the availability of P and N by the presence of LN.

## 4. Conclusions

Nanoparticles composed of physiological lipids (LN) have been proposed as prominent nanocarriers for LC delivery in agriculture practices, being successfully produced with a mean particle size of 165.4 ± 2.343 nm, narrow size distribution, and good physical stability for at least 4 months, at 25 °C. In addition, the high %EE values (98.44 ± 0.04%) indicate the excellent compatibility of LC and the lipid matrix of LN.

Regarding the environmental safety studies, LN showed no risk to the growth of *S. lycopersicum* and *Z. mays* species, as well as to the survival and reproduction of the soil invertebrates *F. candida* and *E. fetida*. However, the earthworms were sensitive to the presence of LN and LN–LC, avoiding its presence at least in the first 48 h of contact. This probably happened due to the lipid composition of both nanoparticles which has favored the interaction with cuticular sensorial cells. This avoiding effect is not necessarily negative, as it may prevent a more intensive contact of earthworms with the encapsulated insecticide; it should be noted that the highest concentration tested for the different formulations was higher than the application dose. Therefore, in the present study, these findings allow us to conclude that LN may be an alternative to encapsulate LC, as no significant effects are expected on soil biota. This was also reinforced by the long-term tests performed. These environmentally safe concentrations of LN–LC need now to be tested for their efficacy on target species of insects.

## Figures and Tables

**Figure 1 nanomaterials-12-02576-f001:**
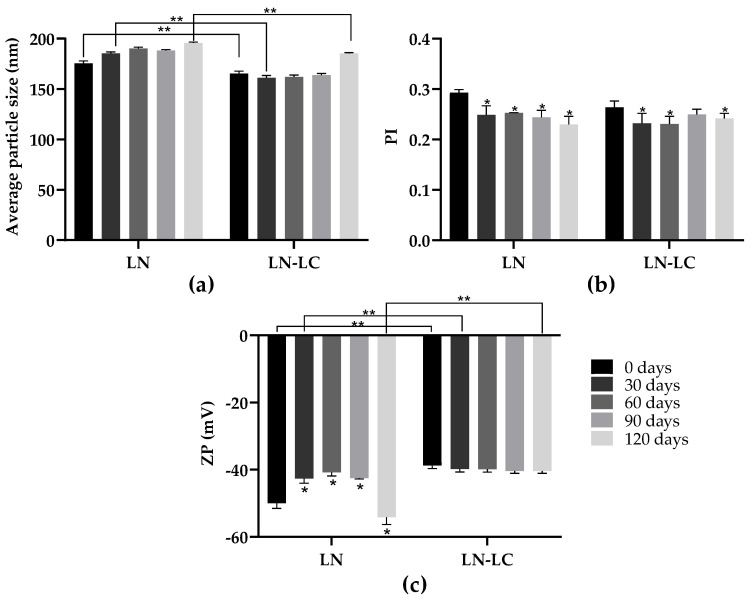
Average particle size (nm) (**A**), polydispersity index (PI) (**B**), and ZP (mV) (**C**) of synthesized LN, monitored at 25 °C immediately after synthesis (0 days) and after 30, 90, and 120 days of storage. Results are expressed as mean ± SD (n = 3). Significant differences between 0 days and other periods for the same LN are represented by (*), *p* < 0.05 (ANOVA followed by Tukey test). Significant differences between unloaded LN and LN loaded with LC for the same period are represented by (**), *p* < 0.05 (two-way ANOVA, followed by Tukey’s test).

**Figure 2 nanomaterials-12-02576-f002:**
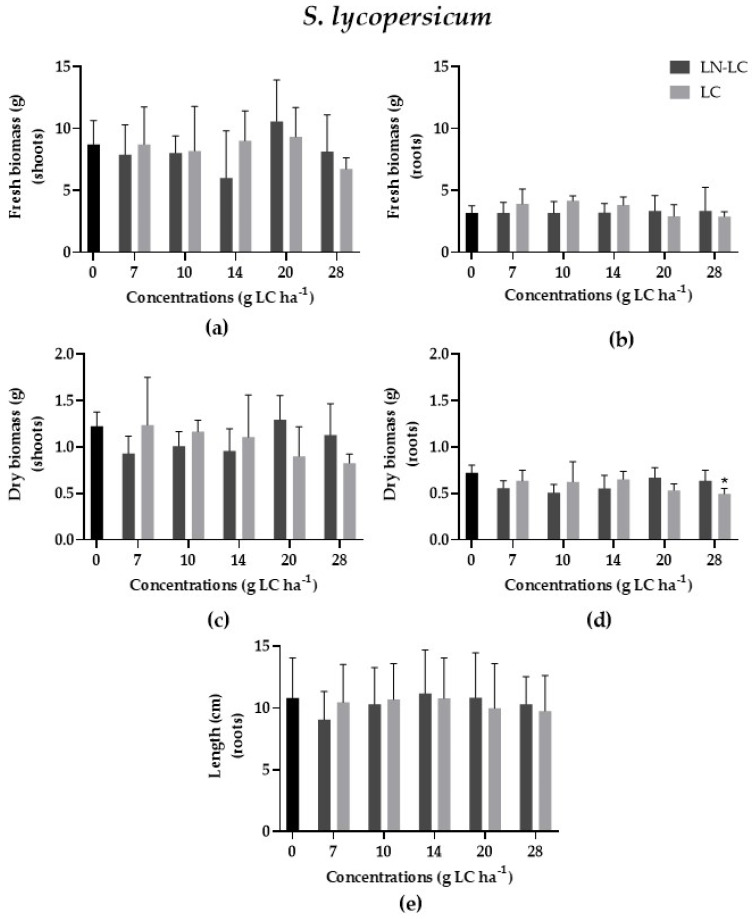
Variation of fresh biomass of shoots and roots (**a**,**b**), dry biomass of shoots, and roots (**c**,**d**) and length (**e**) of roots of *S. lycopersicum* after exposure to LC (light) and LN–LC (dark). The concentrations tested are based on the amount of LC. Results are represented by mean ± SD values. Asterisks mark the significant differences compared to the control group (0 g LC ha^−1^) ((**a**,**b**) *p* < 0.05, Dunnett; (**c**–**e**) *p* < 0.05, Dunn’s).

**Figure 3 nanomaterials-12-02576-f003:**
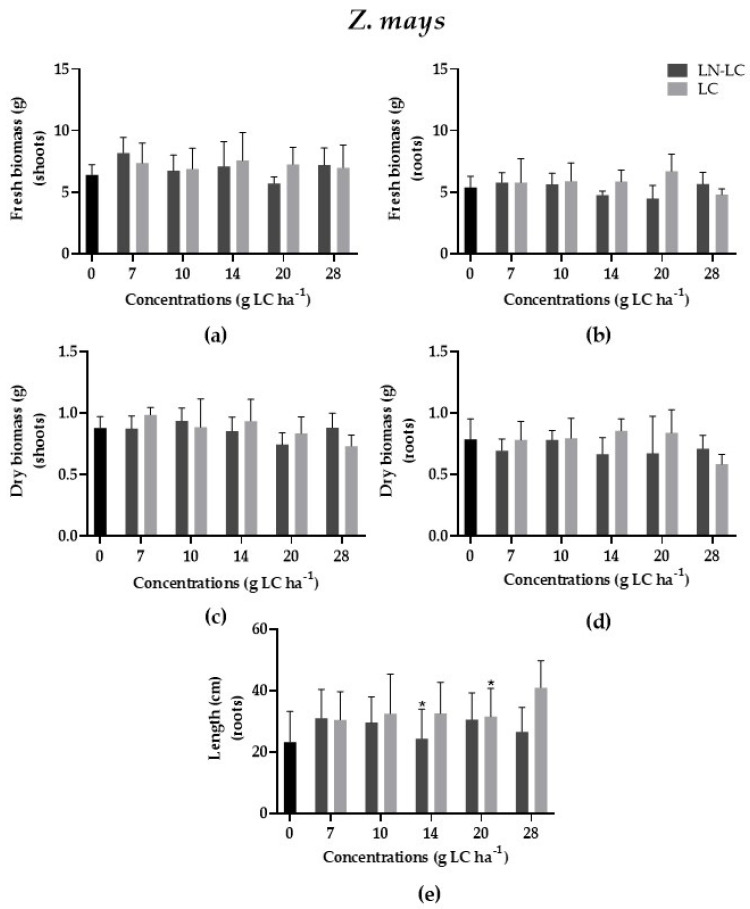
Variation of fresh biomass of shoots and roots (**a**,**b**), dry biomass of shoots and roots (**c**,**d**), and length (**e**) of roots of *Z. mays* after exposure to LC (light) and LN–LC (dark). The concentrations tested are based on the amount of LC. Results are represented by mean ± SD values. Asterisks mark the significant differences compared to the control group (0 g LC ha^−1^): (**a**,**c**,**e**) *p* < 0.05, Dunnett; (**b**,**d**) *p* < 0.05, Dunn’s.

**Figure 4 nanomaterials-12-02576-f004:**
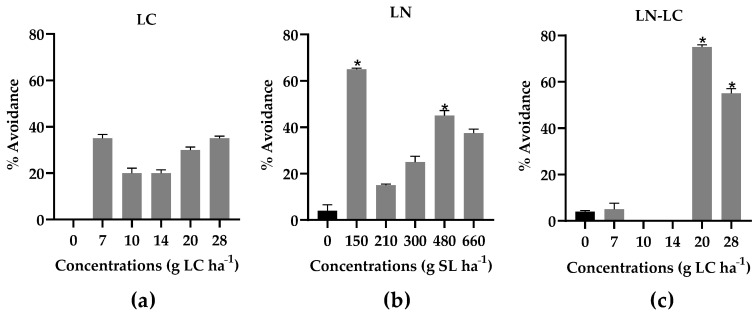
Percentage avoidance of *E. fetida* after exposure to different concentrations of LC (**a**), LN, (**b**) and LN–LC (**c**). The concentrations tested are based on the amount of LC. In the case of LN, the concentrations indicated are of solid lipid (SL) used in the synthesis of LN. The same concentrations of SL were tested for LN–LC. Results are represented by mean ± SD values. Asterisks mark significant differences compared to the control group (0 g LC or LS ha^−1^) (*p* < 0.05, Fisher´s test).

**Figure 5 nanomaterials-12-02576-f005:**
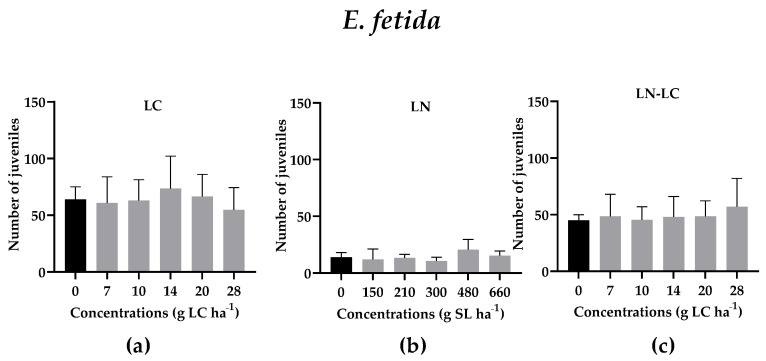
*E. fetida* juveniles after the exposure of adults to different concentrations of LC (**a**), LN (**b**) and LN–LC (**c**). The concentrations tested are based on the amount of LC. In the case of LN, the concentrations indicated are of solid lipid (SL) used in the synthesis of LN. The same concentrations of SL were tested for LN–LC. Results are represented by mean ± SD values (*p* < 0.05, Dunnett).

**Figure 6 nanomaterials-12-02576-f006:**
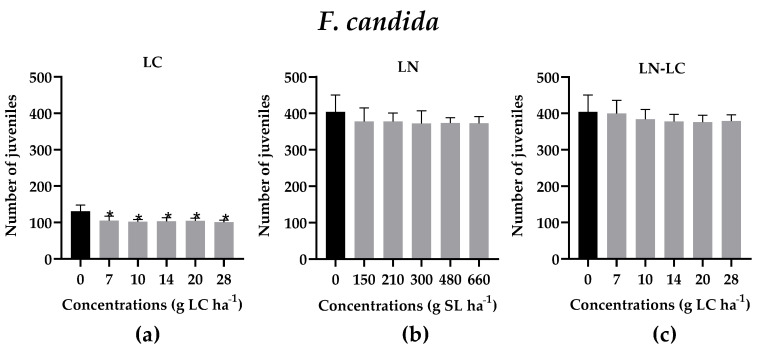
*F. candida* juveniles after the exposure of adults to different concentrations of LC (**a**), LN (**b**), and LN–LC (**c**). The concentrations tested are based on the concentration of LC in the commercial formulation. In the case of LN, the concentrations indicated are of solid lipid (SL) used in the synthesis of LN. The same concentrations of SL were tested for LN–LC. Results are represented by mean ± SD values. Asterisks mark the significant differences compared to the control group (0 g LC ha^−1^) (*p* < 0.05, Dunnett).

**Figure 7 nanomaterials-12-02576-f007:**
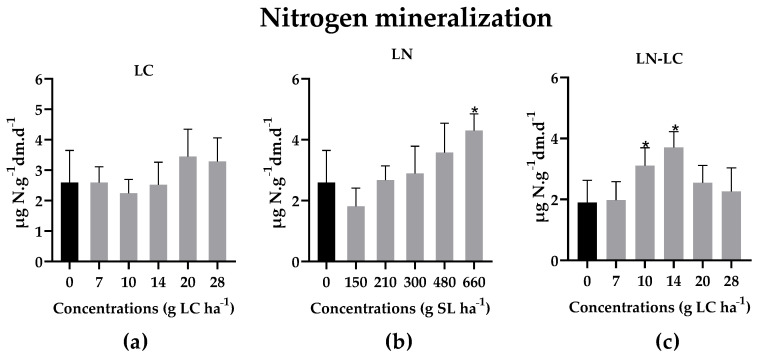
Nitrogen mineralization activity in soils exposed for 15 days to different concentrations of LC (**a**), LN (**b**), and LN–LC (**c**). The concentrations tested are based on the amount of LC. In the case of LN, the concentrations indicated are of solid lipid (SL) used in the synthesis of LN. The same concentrations of SL were tested for LN–LC. Results are represented by mean ± SD values. Asterisks mark the significant differences in relation to the control group (0 g LC ha^−1^): (**c**) *p* < 0.05, Dunnett; (**a**,**b**) *p* < 0.05, Dunn’s.

**Figure 8 nanomaterials-12-02576-f008:**
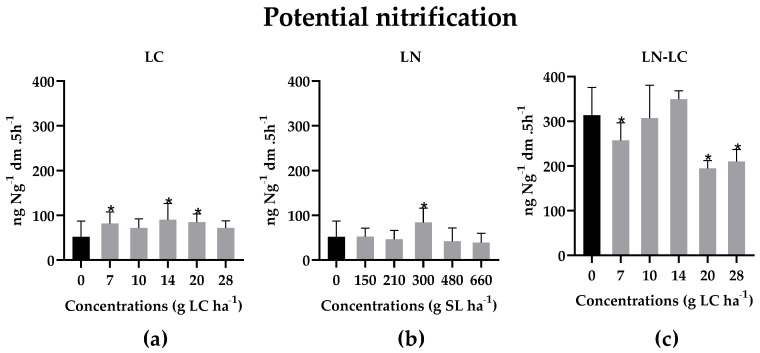
Nitrification potential activity in soils exposed for 15 days to different concentrations of LC (**a**), LN (**b**), and LN–LC (**c**). The concentrations tested are based on the amount of LC. In the case of LN, the concentrations indicated are of solid lipid (SL) used in the synthesis of LN. The same concentrations of SL were tested for LN–LC. Results are represented by mean ± SD values. Asterisks mark the significant differences in relation to the control group (0 g LC ha^−1^) (*p* < 0,05, Dunnett).

**Table 1 nanomaterials-12-02576-t001:** Physicochemical characterization (mean values ± standard deviation (SD)) of the natural soil.

**Natural Soil**	**pH** **(H_2_O)**	**pH** **(KCl)**	**Conductivity (mS cm** **^−1^)**	**% OM**	**% WHC_max_**
	6.46 ± 0.020	5.39 ± 0.32	0.77 ± 0.045	5.21 ± 0.15	43.00 ± 0.15

% OM, percentage of organic matter; % WHCmax, water-holding capacity in percentage of dry mass.

**Table 2 nanomaterials-12-02576-t002:** Average pH values determined for the natural soil containing the different formulations of lambda-cyhalothrin (LC) at the concentration of 28 g LC ha^−1^ after 15 days of incubation. Results are represented by mean values ± standard deviation (SD) (n = 3).

**pH (KCl, 1M)**	**pH Values for the Natural Soil**		
	**Control soil**	**Soil containing LC**	**Soil containing LN**
**pH (KCl, 1M)**	4.59 ± 0.006	4.61 ± 0.010	4.64 * ± 0.006
	**Control soil**	**Soil containing LN–LC**	
	4.69 ± 0.006	4.67 * ± 0.012	

* Significance α = 0.05, compared with respective control soil.

**Table 3 nanomaterials-12-02576-t003:** Soil microbial parameters. Enzymatic activity of dehydrogenase, CM-cellulase, urease, arylsulfatase, and acid-phosphatase after exposure to different concentration of LC, LN, and LN–LC. The concentrations tested are based on the amount of LC. In the case of LN, the concentrations indicated are of solid lipid (SL) used in the synthesis of LN. Results are represented by mean ± SD values. The same concentrations of SL were tested for LN–LC. Asterisks mark the significant differences in relation to the control group (0 g LC ha^−1^).

Soil Microbial Parameters						
		Dehydrogenase (µg TPF g^−1^dm h^−1^)	CM-Cellulase (µg GLU g^−1^dm 24 h^−1^)	Urease (µg NH_4_^+^ g^−1^dm 2 h^−1^)	Arylsulfatase (µg pNP g^−1^dm h^−1^)	Acid Phosphatase (µg pNP g^−1^dm h^−1^)
**LC (g LC ha^−1^)**	**0**	0.71 ± 0.24	107.90 ± 33.28	0.99 ± 0.74	84.57 ± 10.89	234.19 ± 22.34
	**7**	0.62 ± 0.24	128.27 ± 64.39	1.08 ± 0.94	88.69 ± 14.70	225.16 ± 10. 99
	**10**	0.82 ± 0.28	116.05 ± 52.04	1.37 ± 0.80	87.04 ± 8.50	231.52 ± 17.59
	**14**	0.65 ± 0.35	105.57 ± 37.94	3.03 ± 1.54	82.87 ± 6.75	158.41 * ± 59.82
	**20**	0.73 ± 0.16	94.83 ± 106.75	1.93 ± 2.22	90.57 ± 20.78	179.46 * ± 47.73
	**28**	0.83 ± 0.24	137.56 ± 31.04	1.85 ± 1.42	87.86 ± 8.00	159.38 * ± 34.37
**LN (g SL ha^−1^)**	**0**	0.71 ± 0.24	107. 90 ± 33.28	0.99 ± 0.74	84.57 ± 10.90	234.19 ± 22.34
	**150**	0.38 ± 0.25	118.48 ± 38.01	1.81 ± 1.14	74.42 ± 7.00	145.39 * ± 38.61
	**210**	1.68 * ± 0.38	82.83 ± 28.80	1.25 ± 0.80	76.17 ± 8.76	142.83 * ± 44.58
	**300**	1.75 * ± 0.36	74.50 ± 35.41	1.62 ± 1.30	76.74 ± 9.60	147.80 * ± 36.68
	**480**	0.48 ± 0.22	103.75 ± 42.01	2.74 * ± 1.34	72.56* ± 11.33	138.05 * ± 52.47
	**660**	0.33 * ± 0.20	136.13 ± 76.44	5.86 * ± 2.90	75.24 ± 7.33	157.91 * ± 45.49
**LN–LC (g LC ha^−1^)**	**0**	2.16 ± 0.83	63.99 ± 27.73	4.92 ± 0.50	109.07 ± 22.63	234.06 ± 18.67
	**7**	1.74 ± 0.62	71.65 ± 48.31	3.22 ± 0.43	92.38 ± 13.04	227.93 ± 14.64
	**10**	1.71 ± 0.64	99.73 ± 50.14	8.03 * ± 2.11	106.05 ± 17.68	229.61 ± 15.81
	**14**	1.71 ± 0.48	149.98 * ± 45.90	8.91 * ± 1.52	114.85 ± 18.86	224.07 ± 12.68
	**20**	1.33 ± 0.58	107.49 ± 45.92	7.61 * ± 1.92	86.41 * ± 12.30	210.37 * ± 18.85
	**28**	1.21 * ± 0.36	121.39 * ± 41.45	6.00 ± 1.20	78.01 * ± 15.53	190.02 * ± 34.93

## Data Availability

Not applicable.

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
