# Peer review of "Environmental Safety Assessments of Lipid Nanoparticles Loaded with Lambda-Cyhalothrin"

_nanomaterials, 2022, doi:10.3390/nano12152576_

Round 1

Reviewer 1 Report

The paper by Canilho et al. shows a range of interesting results related to the environmental safety assessment of lipid nanoparticles which could potentially be used as carriers for active ingredients in plant protection products. In their study they have included lamba-cyhalotrhin as an example of an AI for which this formulation technique to be performed.

The authors are commended for a range of well-conducted experiments on plants, soil invertebrates, and microbial endpoints in soil as well as their proof of successful … of stable and well-characterized LN. I agree on the conclusions reached for LN and the LN-LC with regards to the outcome of the ecotoxicological studies and as such this work is solid and scientifically sound. However, I have a couple of concerns that must be addressed for the paper to be published in Nanomaterials:

1.      One of my major concern is actually already expressed by the authors in their last sentence in the conclusion: “These environmentally safe concentrations of LN-LC need now to be tested for its efficacy 684 on target species of insects.”. When reading the paper, I was always struggling with the question whether the AI (lamba-cyhalothrin) was actually bioavailable when encapsulated in the LN, i.e. whether the AI kept its insecticidal action intact. If this is not the case and the efficiency of products using LN encapsulation in significantly reduced, the encapsulation in meaningless. This must be addressed by the authors and preferably by tests with target organisms.

2.      The authors mention the “safety-by-design” approach in the abstract and in the introduction as a driver for the work carried out. It is evident that the SbD will be of high importance in the coming years but, related to my first concern, SbD is always accompanied by the question of whether the chemical or product keep its original function and efficiency. For a highly hazardous and highly lipohilic compound like lamba-cyhalothrin, there can of course be a lot gained by better dispersion and slow release but before any claims on SbD can be made, the environmental residence time of LN encapsulated lamba-cyhalothrin must be considered. It will not be safer, if the residence time is increased and abiotic/biological degradation is decreased (as referred to in line 406-408). Actually, I believe that the SbD parts of the paper should be deleted to avoid any confusion in this direction.

3.      The choice of lamba-cyhalothrin as the study compound needs further explanation. Though I am aware of the continued wide use of this AI, it is striking that the authors do not mention the regulatory status of the compound in Europe, i.e. re-authorization was not granted and that all products containing lambda-cyhalothrin should be retracted from the European marked no later than July 20, 2021. The risk for non-target arthropods weighted highly in this decision and therefore the future of lamba-cyhalothrin relies on studies showing that the risk towards these can be adequately managed. Here LN encapsulation could hold some promises but since the present study is not focused at non-target arthropods, this can of course not be claimed. In short, the authors must related to the compound chosen since the regulatory frame is a driver for the development of potentially more safe plant protection products using LN encapsulation.

A few minor concerns:

The paper appear a bit too long compared to the actual results gained. I would suggest to:

1.      shorten section 2.2,

2.      delete Table 2 and to shorten/delete the text on the pH changes in lines 528-532 (the pH is stable 4.6 and 4.7, respectively, and the minute changes observed are of no physiological relevance for the microbial community),

3.      significantly shorten the text on enzymatic endpoint in line 537-630 including the replacement of Fig  7-11 with a table (move Figs to SI)

4.      consider carefully whether all 79 references are indeed needed – for a paper like this, I would think that around 40 references would be sufficient.

Author Response

Dear Reviewer

We would like to say that we are very grateful for the careful reading and revision  aimed at improving our manuscript. We appreciate your time and the detailed valuable comments which were attended and improved the quality of the manuscript. Answers are given below, and annotated version of the manuscript and Supplementary Materials endorsed together with the answers to reviewers. English language and style were also checked.

Response to Reviewer 1

The paper by Ganilho et al. shows a range of interesting results related to the environmental safety assessment of lipid nanoparticles which could potentially be used as carriers for active ingredients in plant protection products. In their study they have included lamba-cyhalothrin as an example of an AI for which this formulation technique to be performed. 

The authors are commended for a range of well-conducted experiments on plants, soil invertebrates, and microbial endpoints in soil as well as their proof of successful of stable and well-characterized LN. I agree on the conclusions reached for LN and the LN-LC with regards to the outcome of the ecotoxicological studies and as such this work is solid and scientifically sound. However, I have a couple of concerns that must be addressed for the paper to be published in Nanomaterials:

  1. One of my major concern is actually already expressed by the authors in their last sentence in the conclusion: “These environmentally safe concentrations of LN-LC need now to be tested for its efficacy 684 on target species of insects.”. When reading the paper, I was always struggling with the question whether the AI (lamba-cyhalothrin) was actually bioavailable when encapsulated in the LN, i.e. whether the AI kept its insecticidal action intact. If this is not the case and the efficiency of products using LN encapsulation in significantly reduced, the encapsulation in meaningless. This must be addressed by the authors and preferably by tests with target organisms.

Authors: We appreciate the suggestion, and we agree with the reviewer. It is true that if the nanoencapsulation is not efficient, the activity of AI may be compromised. However, our objective in the present study was to first obtain enough data for risk assessment for soil and terrestrial organism and then test the encapsulated AI on target organism as mentioned. This will help us to understand if the AI (lambda-cyhalothrin) is bioavailable maintaining insecticidal action following toxicity procedures on model insect species like Drosophila melanogaster. The assays with target species are already being carried out. Since we believe that lipid nanoparticles (LN) have a great potential for encapsulation of this and other AI, we considered that it was important to start testing intrinsic toxicity of LN, as well as of the encapsulated AI to non-target species. In fact, and in the other way around, the AI was removed from the market (as the reviewer remarked), and if we found that once encapsulated it becomes even more toxic, it thus not makes sense to proceed with the evaluation of its efficiency.   

  1. The authors mention the “safety-by-design” approach in the abstract and in the introduction as a driver for the work carried out. It is evident that the SbD will be of high importance in the coming years but, related to my first concern, SbD is always accompanied by the question of whether the chemical or product keep its original function and efficiency. For a highly hazardous and highly lipophilic compound like lamba-cyhalothrin, there can of course be a lot gained by better dispersion and slow release but before any claims on SbD can be made, the environmental residence time of LN encapsulated lamba-cyhalothrin must be considered. It will not be safer, if the residence time is increased and abiotic/biological degradation is decreased (as referred to in line 406-408). Actually, I believe that the SbD parts of the paper should be deleted to avoid any confusion in this direction.

Authors: We appreciate the suggestion and comments from reviewer. Although the nanoencapsulation can lead to an increase of residence time of pesticides, we believe that the encapsulation of lambda-cyhalothrin in physiological/biodegradable lipids may increase the degradation by microbial community, as lipid can serve as substrate for the soil microbial growth and activity by inducing the production of microbial enzymes responsible for catabolic reactions. The SbD concept is “about including safety at the earliest possible stage of product and process development” (https://safe-by-design-nl.nl/home+english/about+safe-by-design/default.asp). Therefore, we consider that is precisely this approach that we are following, as the first approach was to test the safety of encapsulated LC to non-target species.

  1. The choice of lamba-cyhalothrin as the study compound needs further explanation. Though I am aware of the continued wide use of this AI, it is striking that the authors do not mention the regulatory status of the compound in Europe, i.e. re-authorization was not granted and that all products containing lambda-cyhalothrin should be retracted from the European marked no later than July 20, 2021. The risk for non-target arthropods weighted highly in this decision and therefore the future of lamba-cyhalothrin relies on studies showing that the risk towards these can be adequately managed. Here LN encapsulation could hold some promises but since the present study is not focused at non-target arthropods, this can of course not be claimed. In short, the authors must related to the compound chosen since the regulatory frame is a driver for the development of potentially more safe plant protection products using LN encapsulation.

Authors: We acknowledge the referee’s remark. We agree with the reviewer and have revised the introduction considering the reviewer suggestion. We added this information in the introduction. Although lambda cyhalothrin (LC) has been banned from the European market, the use of this insecticide is still authorized in other countries and there still are several studies focusing on the toxicity evaluation of LC (https://doi.org/10.1016/j.jhazmat.2021.126853) and on the development of sustainable insecticide formulations, using for example, the encapsulation of LC to improve the insecticide efficiency(https://doi.org/10.3390/nano11102730)and(https://doi.org/10.1016/j.scitotenv.2022.154914). In this sense, the nanoencapsulation of LC in physiological lipid nanoparticles can be a promising alternative to decrease the harmful effects of these compounds. As we mentioned above, we hypothesize that LN, can control the release of AI, but also to promote the degradation of the residues in soils and water, as LN can act as a source of carbon to microorganisms. In addition, up-to-date, and despite this AI has been banned, studies based on the toxicity effects of LC on non-target species, especially terrestrial organisms are still limited and thus, we believe that the information contained in our work may improve the understanding of the environmental safety assessment of LC.

A few minor concerns:

The paper appear a bit too long compared to the actual results gained. I would suggest to:

  1. shorten section 2.2, 

Authors: We appreciate the reviewer suggestion. The section 2.2 was shortened accordingly, as well as the results and discussion sections.

  1. delete Table 2 and to shorten/delete the text on the pH changes in lines 528-532 (the pH is stable 4.6 and 4.7, respectively, and the minute changes observed are of no physiological relevance for the microbial community),

      Authors: We appreciate the suggestion, but although the changes are not really very large, they may in fact have had some influence. Thus, we decided to keep it, because although the variations were small, they were significant.

  1. significantly shorten the text on enzymatic endpoint in line 537-630 including the replacement of Fig 7-11 with a table (move Figs to SI),

Authors: The replacement of Fig 7-11 by Table was made according to the reviewer suggestion. However, the text of enzymatic data was maintained, since we believe that such information is crucial for the best understanding of the results.

  1. consider carefully whether all 79 references are indeed needed – for a paper like this, I would think that around 40 references would be sufficient.

Authors: We appreciate the valuable suggestions of the reviewer. We revise the references according to reviewer suggestion.

Reviewer 2 Report

The manuscript entitled “Environmental safety assessments of lipid nanoparticles loaded 2 with lambda-cyhalothrin” is well written and of interest to the Journal’s readers.  It referes to the encapsulation of some pesticides in lipid nanoparticles to decrease the environmental toxicity. The authors claim that the insecticedes thusconditioned are less harmful to non-target species. Several measurements asure that the soil is sohow protected by this new echnology. Therefore, I consider that the manuscript has enough merits to be published.

Author Response

Dear reviewer

We would like to say that we are very grateful for the careful reading and revision  of the manuscript. We appreciate your time and the valuable comments. 

Response to Reviewer 2

The manuscript entitled “Environmental safety assessments of lipid nanoparticles loaded 2 with lambda-cyhalothrin” is well written and of interest to the Journal’s readers.  It refers to the encapsulation of some pesticides in lipid nanoparticles to decrease the environmental toxicity. The authors claim that the insecticides thus conditioned are less harmful to non-target species. Several measurements asure that the soil is sohow protected by this new echnology. Therefore, I consider that the manuscript has enough merits to be published.

Authors: We appreciate the valuable comments of reviewer on our manuscript.